# Are Two Heads the Same as One? Identifying Disparate Treatment in Fair Neural Networks

**Michael Lohaus**[*]
University of Tübingen
Tübingen, Germany
mlohaus@yahoo.com

**Matthäus Kleindessner**
Amazon Web Services
Tübingen, Germany
matkle@amazon.com

**Krishnaram Kenthapadi**
Fiddler AI
Palo Alto, USA
krishnaram@fiddler.ai

**Francesco Locatello**
Amazon Web Services
Tübingen, Germany
locatelf@amazon.com

**Chris Russell**
Amazon Web Services
Tübingen, Germany
cmruss@amazon.com

## Abstract

We show that deep networks trained to satisfy demographic parity often do so through a form of race or gender awareness, and that the more we force a network to be fair, the more accurately we can recover race or gender from the internal state of the network. Based on this observation, we investigate an alternative fairness approach: we add a second classification head to the network to explicitly predict the protected attribute (such as race or gender) alongside the original task. After training the two-headed network, we enforce demographic parity by merging the two heads, creating a network with the same architecture as the original network. We establish a close relationship between existing approaches and our approach by showing (1) that the decisions of a fair classifier are well-approximated by our approach, and (2) that an unfair and optimally accurate classifier can be recovered from a fair classifier and our second head predicting the protected attribute. We use our explicit formulation to argue that the existing fairness approaches, just as ours, demonstrate *disparate treatment* and that they are likely to be unlawful in a wide range of scenarios under US law.

## 1 Introduction

Autonomous systems that make substantive decisions about people must conform to relevant anti-discrimination legislation. Within the US legal system, two common tests of anti-discrimination legislation are referred to as disparate treatment and disparate impact [37]. Disparate treatment corresponds to the idea that people should not be treated differently because of their membership of a protected group (such as race or gender), while disparate impact refers to the idea that seemingly neutral practices should not cause a substantial adverse impact to a protected group. Importantly, it has been argued [7] that there are a large range of scenarios where disparate treatment is unlawful even when performed as a remedy to disparate impact. This is in departure from the EU where more latitude exists when rectifying indirect discrimination (analogous to disparate impact) [67].

Consequentially, disparate treatment doctrine prevents a wide range of actions intended to address sustained inequality [9]. Of particular relevance to our work is a 1991 amendment to Title VII [60]. This amendment explicitly prohibits the "use [of] different cutoff scores for . . . employment related tests on the basis of race, color, religion, sex, or national origin", even if done for reasons of

---

[*]Work done during an internship at Amazon Web Services.

36th Conference on Neural Information Processing Systems (NeurIPS 2022).

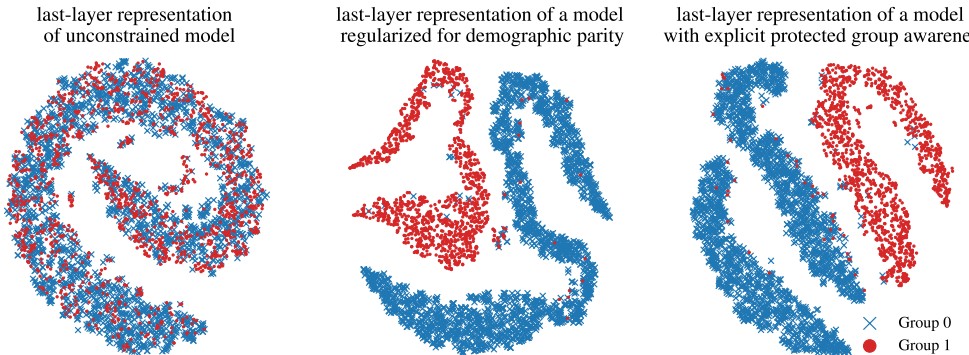

last-layer representation of unconstrained model

last-layer representation of a model regularized for demographic parity

last-layer representation of a model with explicit protected group awareness

Group 0
Group 1

Figure 1: **Feature representations of unconstrained (*left*), fairness-regularized (*center*), and group-aware (*right*) ResNet50 models.** The plots show tSNE [65] embeddings of last-layer representations of the CelebA dataset [43]; red points correspond to male individuals, blue points to female ones. Classifiers are trained to identify if people are smiling. The use of fairness constraints during training causes a mixed manifold (*left*) to separate into largely disjoint sets (*center*). Similar behavior is observed when training a two-headed model (*right*) that predicts gender and 'smiling'.

affirmative action. In this paper, we examine the relationship between existing methods for enforcing demographic parity (a definition closely related to avoiding disparate impact) that are commonly claimed to not exhibit disparate treatment and methods for demographic parity that alter the cutoff score on the basis of inferred race or gender. In doing so, we raise fundamental questions about the legality of existing approaches for enforcing demographic parity.

In particular, we examine the behavior of deep neural networks trained to satisfy demographic parity either with a fairness regularizer [72] or by preprocessing [33]. Such models could learn an internal representation that either *(i)* is independent of the protected attribute (as methods from fair representation learning aim to do—see Appendix B for related work), or *(ii)* that distinguishes between groups so as to tune "separate" classifiers for each group in a way that results in the demographic parity of the network. In the context of US law, it is vital to understand which case occurs in practice. If the learned algorithms treat people differently on the basis of their race or gender, this may correspond to *disparate treatment*. We find that networks trained to satisfy demographic parity via a fairness regularizer or preprocessing fall into the second case. Figure 1 provides an illustration of this finding. Moreover, we find that the more strongly demographic parity is enforced, the more predictive the internal representation is of the protected attribute.

Building on the observation that the protected attribute is implicitly learned, we train a neural network with a second classification head that explicitly predicts the protected attribute. After training, we use the second head response to directly reduce demographic disparity. On certain tasks we show that our approach is tightly related to existing regularizer and preprocessing approaches and conclude that these approaches, just as ours, demonstrate disparate treatment.

To summarize, we make the following contributions:

1. The internal state of a fair network trained with a fairness regularizer or preprocessing is strongly predictive of the protected attribute. As the fairness is more strongly enforced, the accuracy of predicting the protected attribute increases (Section 3).

2. Merging a second classification head with the network is an effective approach for creating demographic parity-fair classifiers. We show that the performance of our approach closely tracks the optimal accuracy-fairness trade-offs of Lipton et al. [42]; however, in contrast to their method, ours does not require explicit access to the protected attribute (Section 4).

3. As our principal contribution, we show cases where the decisions of a fair model (via regularizer or preprocessing) are well-approximated by our approach. Similarly, the decisions of an unconstrained classifier can be approximated by a weighted sum of the fair classifier and our second head (Section 5.1).

4. Using the close relationship between a demographic parity-fair classifier and our explicit approach we are able to identify individuals who are systematically disadvantaged by the fair classifier, and

who would receive a positive decision if their apparent race or gender were different. This allows us to conclude that the fair method exhibits disparate treatment (Section 5.2).

## 2 Preliminaries

A binary classifier is a function $h : \mathcal{X} \to \mathcal{Y}$ that, given a datapoint $x$ from feature space $\mathcal{X}$, aims to accurately predict the datapoint's assigned label $y \in \mathcal{Y} = \{0, 1\}$. We consider thresholded classifiers of the form $h(x) = \mathbb{1}(f(x) > 0)$, where $f$ is a continuous function $f : \mathcal{X} \to \mathbb{R}$. We want $h$ to be as accurate as possible, while at the same time being fair with respect to a protected attribute $s \in \mathcal{S} = \{0, 1\}$ according to the fairness notion of demographic parity.

**Definition 2.1 (Demographic Parity)** *[33, 18] A classifier $h : \mathcal{X} \to \{0, 1\}$ satisfies demographic parity under distribution $\mathbb{P}$ if its prediction $h(X)$ is independent of the protected attribute $S$, i.e.,*

$$\mathbb{P}(h(X) = 1 | S = 0) = \mathbb{P}(h(X) = 1 | S = 1).$$

*We measure the violation of demographic parity by the demographic disparity (DDP), given by*

$$\mathrm{DDP} = \mathrm{DDP}(h; \mathbb{P}) := \mathbb{P}(h(X) = 1 | S = 1) - \mathbb{P}(h(X) = 1 | S = 0). \tag{1}$$

In this paper, we examine two popular fairness methods for training deep neural networks to satisfy demographic parity:

**Regularized Fair Classification** One common approach for learning a fair classifier is to add a regularizer to the standard training objective [e.g., 11, 72, 47, 44, 40, 56, 8]. Such regularizers are used to enforce numerous fairness definitions and typically impose a continuous relaxation of a discrete fairness measure such as the DDP (1). The regularizer used in [72] is a sigmoid-based relaxation of the squared value of (1), evaluated on a dataset $(x_i, y_i, s_i)_{i=1}^{n}$ with i.i.d. samples from $\mathbb{P}$:

$$\widehat{\mathcal{R}}_{\mathrm{DP}}(f) := \left( \frac{1}{|\{i : s_i = 1\}|} \sum_{i \in [n]: s_i = 1} \sigma(f(x_i)) - \frac{1}{|\{i : s_i = 0\}|} \sum_{i \in [n]: s_i = 0} \sigma(f(x_i)) \right)^2. \tag{2}$$

The fairness regularizer is added to the overall loss and trades-off fairness versus accuracy via a multiplicative hyperparameter $\lambda$, where higher values of lambda encourage an increase in fairness at the cost of accuracy. In the paper, we report experimental results for $\widehat{\mathcal{R}}_{\mathrm{DP}}(f)$. Appendix D presents results for a related regularizer. Note that we follow [8, 72] in applying the regularizer (2) to the sigmoid output of the networks. This differs from approaches such as [76, 16] that enforce relaxed fairness constraints on margin distances and have recently been criticized for their ineffectiveness [44].

**Preprocessing: Massaging the Dataset** We also examine the preprocessing method of [33], which alters target labels prior to training. *Massaging* 'promotes' negative points from the disadvantaged group to the positive label if they have a (comparatively) high positive class probability as calculated by an unconstrained classifier and 'demotes' positive points from the advantaged group to the negative label if they have a low positive class probability. The number of changed labels is controlled by a parameter $\lambda \in [0, M]$, where $\lambda = M$ results in the same fraction of positive points for both groups.

**Fair Representation Learning** methods learn data representations such that any ML model trained on top of such a representation would be fair [78, 46, 10, 1, 79, 20, 45, 23, 50, 75, 30, 54, 5]. These techniques come in various flavors, and are considered preprocessing or in-processing methods. When used to train demographic parity-fair classifiers, these methods try to learn representations that contain no information of the protected attribute Consequently, these techniques should not show the phenomenon of attribute awareness we found for networks regularized to satisfy demographic parity or trained on massaged datasets, at least not when examining the final representation layer. However, it is an interesting question for future work whether methods for fair representation learning suffer from attribute awareness in earlier network layers. Like us, adversarial approaches like [46, 10, 1, 20, 20] train a second head, but in contrast to our approach they are trained adversarially in a minmax formulation, while the objectives of our two heads are jointly minimized in a multitask setting.

In general these fair representation learning methods have noticeably different behavior to preprocessing or regularized fairness methods. Because they try to solve a harder problem of learning a representation that guarantees the fairness of any downstream classifier, rather than making a single

classifier fair, they typically show worse accuracy trade-offs than other methods. As such, while we did not did not attempt the analysis set out in Section 5 on these methods, we do not expect our fairness results to hold for them.

To set the ground for our paper, we now discuss the closely related work of Lipton et al. [42]. We discuss further related work in Appendix A and B.

**Implicit Disparate Treatment** Lipton et al. [42] examine the popular claim that machine learning models that do not use protected information at test time do not exhibit disparate treatment [24, 77, 76, 16, 74, 47, 26]. Lipton et al. recommend caution and observe that if the protected attribute $s$ is a deterministic function $s = g(x)$ of the non-protected features $x$, any sufficiently powerful ML model can learn a function $f(x, s) = \tilde{f}(x)$ with $\tilde{f}(x) = f(x, g(x))$. They argue that even though the protected attribute is not provided at test time, such a model would constitute a case of disparate treatment since it makes decisions based on the *implicitly reconstructed* protected attribute. However, beyond a synthetic experiment in which a classifier discriminates based on hair length (as a proxy for gender), they do not study whether—or *how*—implicit disparate treatment happens in practice. We provide strong evidence that deep neural networks that are enforced to satisfy demographic parity by means of a regularizer or preprocessing suffer from disparate treatment, even when not explicitly using the protected attribute at test time—and that they do so by separating last-layer representations based on protected attributes.

A second contribution of Lipton et al. [42] is to prove that the most accurate classifier among all classifiers satisfying a bound on the demographic disparity utilizes group-specific thresholds (a similar result has also been shown by Menon and Williamson [49]). They then propose a postprocessing method that greedily chooses per-group decision thresholds for a classifier that has been learned without fairness constraints; this requires access to protected attributes at test time. In Section 4 we compare our proposed two-head approach for demographic parity-fairness to their approach.

## 3  Protected Attribute Awareness in Fair Networks

Here, we provide evidence that deep networks that use a fairness regularizer or preprocessing to satisfy demographic parity, learn an internal representation that separates groups, thus allowing each group to be treated differently. To measure how well a fair network separates the protected groups, we examine how accurately a linear classifier can recover the protected attribute from the output of the last layer. We present results for the fairness regularizer $\widehat{\mathcal{R}}_{\mathrm{DP}}(f)$ defined in (2) on one dataset; see Appendix D.1 for the results with *Massaging* [33], another regularizer and on a second dataset[2].

**Experimental Setup** From the CelebA image dataset [43] and nine of its 40 binary attributes, we choose a target attribute and a protected attribute, such as SMILING and YOUNG. We use [55] as reference for restricting our study to only nine of the 40 attributes. For each distinct pair of target and protected attribute, we train ResNet50 models for twelve different values of the fairness parameter $\lambda$. For each model, we then train a linear classifier using logistic regression to predict the protected attribute from the model's frozen last-layer representation. We refer to these classifiers as the group classifiers. See Appendix C.1 for technical details and descriptions of the datasets.

**Evaluation** For each pair of target and protected attribute, we evaluate if increasing $\lambda$ increases the accuracy of the group classifier. We test for a monotonic relationship using the Kendall-tau correlation $\tau$ [35] on 12 datapoint pairs (consisting of fairness parameter $\lambda$ and the accuracy of the group classifier). Additionally, we compute a two-sided p-value for the null hypothesis of independence between $\lambda$ and the group classifier's accuracy. Since the regularized approach can collapse to a trivial near-constant classifier when $\lambda$ is too large, we discard models when their accuracy is too close to the one of the constant classifier (concretely, if their accuracy is not at least the constant classifier's accuracy plus 25% of the additional accuracy that the unconstrained classifier achieves).

**Results** Figure 2 summarizes our results. For 71 out of 72 experiments the Kendall-tau correlation shows a positive correlation between $\lambda$ and the protected attribute accuracy, with $p < 0.05$ for 62 experiments. Those experiments show a very strong monotonic relationship with a correlation higher than $0.49$. For nine experiments, we observe lower but still positive values of the correlation coefficient $\tau$. Only for one experiment we observe $\tau < 0$, but with insignificant $p = 0.64$.

---

[2]Code is available at `https://github.com/mlohaus/disparatetreatment`.

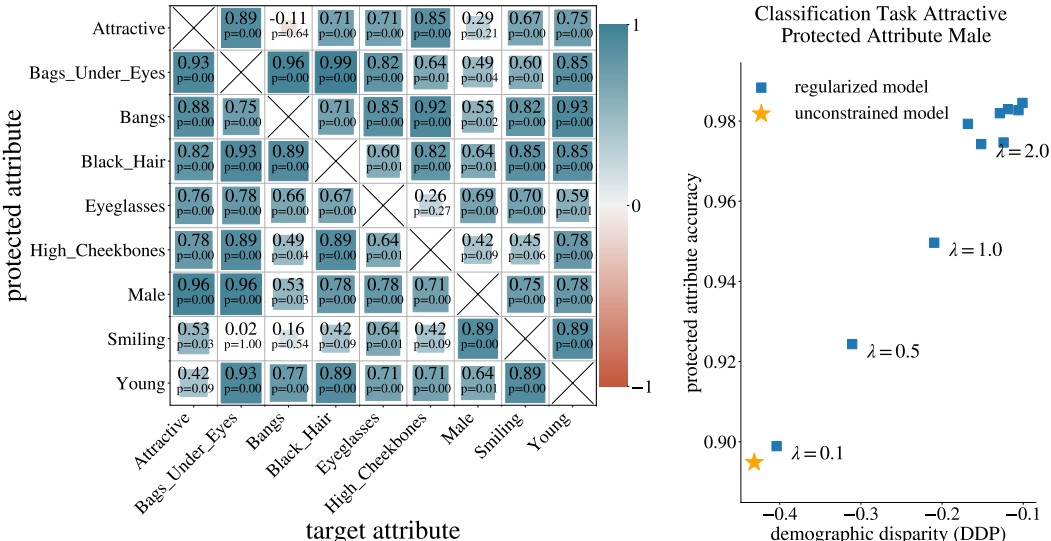

Figure 2: *Left:* **Kendall-tau rank correlations between fairness parameter $\lambda$ and the accuracy of the group classifier** For each pair of protected and target attribute, we first train $12$ regularized ResNet50 models with varying fairness parameter $\lambda$. Subsequently, the linear group classifier is trained on the frozen last-layer representation to predict the protected attribute. From the resulting $12$ pairs of $\lambda$ and protected attribute accuracy, we test for a monotonic relationship between $\lambda$ and protected attribute accuracy by computing the Kendall-tau rank correlation. The color and size of a square correspond to the value and magnitude of the correlation coefficient. **For almost all tasks, the accuracy of predicting the protected attribute increases as the fairness parameter increases.** *Right:* We show how the linear separability of gender increases with the regularization strength when the target attribute is ATTRACTIVE and the protected attribute is MALE. As we increase the fairness parameter $\lambda$, the group classifier accuracy increases by up to more than $8\%$, reaching 98% accuracy.

**Conclusion** For both fairness regularizers and the *Massaging* preprocessing method we find that **increasing the fairness parameter increases the ability to recover the protected attribute from the last layer.** This increase in accuracy is a cause for concern since disparate treatment can occur if a system is able to infer the protected attribute (cf. Section 2). We build on this initial analysis in Sections 4 and 5 and show that disparate treatment occurs in practice.

## 4 Explicit Group Awareness for Enforcing Demographic Parity

In this section, we evaluate a novel post-processing approach **explicitly designed to exhibit disparate treatment** while using the same single network architecture used by preprocessing and regularized approaches. In short, a network with two fully-connected output heads on the last-layer representation is trained: one head $f : \mathcal{X} \to \mathbb{R}$ is trained to minimize a logistic loss over the target variable $y$, and the other head $g : \mathcal{X} \to \mathbb{R}$ to minimize a squared loss over the protected attribute $s$. We use the squared loss to encourage $g$ to take values close to zero or one. Overall, we minimize the following objective:

$$\widehat{L}(f,g) = \frac{1}{n}\left(\sum_{i=1}^{n}\widehat{L}_{\mathrm{BCE}}(\sigma(f(x_i)), y_i) + (g(x_i) - s_i)^2\right).$$

A weighted sum of the two heads results in a classifier $F$ that satisfies (approximate) demographic parity. The function $F$ takes the form $F(x) = f(x) + a_1 g(x) + a_2$ for coefficients $a_1, a_2 \in \mathbb{R}$. Even though we train two heads, we can compress them into a single head and generate *a single-headed architecture that makes the same decisions*. More precisely if $f(x) = w_f \cdot z(x) + b_f$ and $g(x) = w_g \cdot z(x) + b_g$, with $z(x)$ being the last-layer representation, then $F(x) = (w_f + a_1 w_g) \cdot z(x) + (b_f + a_1 b_g + a_2)$. This results in a single-headed network with identical architecture to the preprocessing and regularized approaches. We consider our approach a post-processing method as it doesn't induce a fair classifier until we compress the two heads.

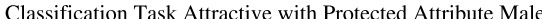

Classification Task Attractive with Protected Attribute Male

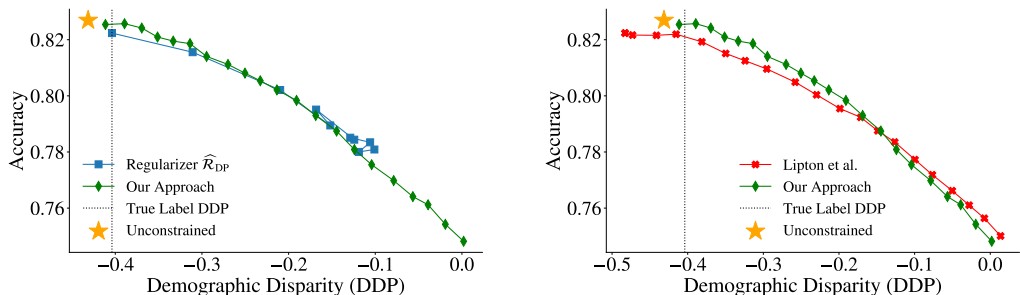

Figure 3: **Comparison of different single architecture approaches** We compare our group aware model to a fairness-regularized model (*left plot*) and the approach of Lipton et al. [42] (*right plot*) on the task of predicting the target ATTRACTIVE with respect to the protected attribute MALE. For all methods, we observe the typical trade-off: as the model becomes fairer (DDP is closer to 0), the target accuracy for ATTRACTIVE decreases. All methods obtain similar accuracy for a particular DDP value. However, the regularizer approach is unable to achieve near perfect fairness and saturates around a DDP value of $-0.1$. Note that Lipton et al. [42] requires the protected attribute at test time, while we infer the protected attribute. More datapoints are shown in the scatter plot for Lipton et al. [42] and our approach because they can be generated by varying thresholds without retraining. In contrast, the regularized approach requires a full-retraining for every choice of fairness parameter.

As we show, there are substantial advantages to post-processing, both in terms of stability and efficiency of training. However, the key limitation of existing methods is that they either require knowledge of the protected attribute [42, 3] or more complex systems at evaluation time that must also infer the protected attribute [73, 51]. We remove this limitation, by compressing our two-headed approach into the same architecture used by the preprocessing and regularized approaches. This matching architecture is key to the legal argument in the following sections: our approach explicitly exhibits disparate treatment while sharing a common architecture and making similar decisions to other approaches.

**Model Construction** We consider a standard network backbone (e.g., ResNet50), with two heads. We present results with a ResNet50 in the body of the paper, and include further results with the MobileNetV3-Small architecture in Appendix D.2. To find $a_1$ and $a_2$ such that the predictions of the thresholding rule $\mathbb{1}(F(x) > 0)$ are fair and maximally accurate, we perform a grid search on validation data and select the most accurate classifier that does not exceed a given demographic disparity.

We compare our approach to Lipton et al. [42] who's similar approach provides optimal per-group thresholds, but requires explicit knowledge of the protected attributes at test time. If the group classifier $g$ is perfect, our approach and Lipton's approach coincide. If $g$ does not predict the protected attribute well, our procedure can only perform worse than Lipton's (cf. Theorem 4 in their paper). In our computer vision setting, however, the accuracy of predicting the protected attribute is typically very high (e.g., on MALE from CelebA, we achieve an accuracy of around $98\%$), and, as we show, the performance of our approach is very close to that of [42].

**Results** Figure 3 compares the accuracy-fairness trade-off of our approach, the regularizer approach, and Lipton et al. [42]. Since the computational cost of fully training 12 regularized models (with the same 12 fairness parameter values as in Section 3) is much higher than training one model for our approach and Lipton's, we compute 20 solutions with our grid search or Lipton's greedy search. All three methods offer similar accuracy for a particular level of demographic disparity. However, for the regularized model, it is difficult to control this trade-off or to find very fair solutions. In contrast, our approach and [42] allow easy selection of a model with a particular demographic disparity. While the performance of our approach and [42] is similar, we do not require the protected attribute at test time.

Table 1 reports the accuracy obtained under strict fairness constraints. With respect to the accuracy-fairness trade-off all methods perform comparably. However, if we require a substantial reduction of unfairness (second block of rows in Table 1), the regularizer approach and preprocessing often fail

Table 1: **Accuracy of single architecture approaches under strict fairness constraints.** We show the accuracy of various approaches at substantially decreased demographic disparity. In the first block, we require the DDP to be half that of the DDP of the unconstrained classifier, in absolute value. For all methods, we report the most accurate model of all sufficiently fair models. In the second, the DDP can be at most 20%. Failure to reach the required fairness is indicated with a cross. For small reductions in disparity, all methods have a similar accuracy-fairness trade-off. However, the regularized approach often fails to find sufficiently fair solutions. **Our approach always finds a sufficiently fair solution** and is comparable to Lipton's approach, which requires the protected attribute at test time.

| | Attractive | Bags_Under_Eyes | Bangs | Black_Hair | Eyeglasses | High_Cheekbones | Smiling | Young |
|---|---|---|---|---|---|---|---|---|
| | | | | 50% disparity reduction | | | | |
| Lipton | 0.7955 | 0.8452 | 0.9469 | 0.8959 | 0.9680 | 0.8641 | 0.9240 | 0.8770 |
| Our Approach | 0.8021 | 0.8419 | 0.9463 | 0.8905 | 0.9775 | 0.8615 | 0.9227 | 0.8756 |
| Massaging | 0.7992 | 0.8337 | 0.9481 | 0.9002 | 0.9718 | 0.8546 | 0.9207 | 0.8679 |
| Regularizer $\widehat{\mathcal{R}}_{\mathrm{DP}}$ | 0.8021 | 0.8274 | ✗ | 0.9023 | ✗ | 0.8350 | ✗ | 0.8701 |
| | | | | 80% disparity reduction | | | | |
| Lipton | 0.7719 | 0.8344 | 0.9336 | 0.8959 | 0.9632 | 0.8456 | 0.9137 | 0.8617 |
| Our Approach | 0.7698 | 0.8332 | 0.9301 | 0.8881 | 0.9647 | 0.8436 | 0.9118 | 0.8592 |
| Massaging | 0.7674 | 0.8185 | ✗ | 0.8989 | ✗ | ✗ | ✗ | 0.8573 |
| Regularizer $\widehat{\mathcal{R}}_{\mathrm{DP}}$ | ✗ | 0.8274 | ✗ | ✗ | ✗ | ✗ | ✗ | ✗ |

to find a valid solution. In contrast, Lipton et al. [42] and our approach always find sufficiently fair solutions due to their direct search for per-group thresholds.

**Conclusion** Our method has several advantages: (i) Other approaches require training a new model for every fairness parameter $\lambda$, which might make tuning $\lambda$ very expensive until a desirable level of fairness is reached. Our approach only requires a single explicit model and the output scores of the two heads. (ii) Due to the simple weighted sum, we make the influence of the group classifier in the final decision explicit and transparent. In summary, **our approach reliably finds high accuracy solutions for a given demographic disparity** without requiring the protected attribute at test time.

## 5    Disparate Treatment in Fair Networks

In this section we examine the tight relationship between our explicit approach and the behavior of fair neural networks. First, we reconstruct the fair network with our group-aware method by building a weighted sum of the two heads and merging them into one (Section 4). Second, from a given fair neural network, we recover the corresponding unconstrained model using only the fair network and the group classifier from our approach. The relationship between fair model and our approach allows us to identify individuals who are treated differently based on inferred group membership and to demonstrate disparate treatment.

### 5.1    Fair Networks Behave like the Explicit Approach

We recover the predictions of a fair model $r_\lambda$ by using the target task head $f$ and group classifier $g$. We use logistic regression to find parameters $a_1, a_2 \in \mathbb{R}$ such that $\mathbb{1}(f(x) + a_1 g(x) + a_2 > 0)$ accurately replicates $\mathbb{1}(r_\lambda(x) > 0)$. Since we can merge the weighted sum into a single output head, our two-headed approach has thus found a model of the same original architecture with one output head that predicts equivalently to the fair model. Next, we show that it is possible to recover the predictions of the unconstrained model $h$ with a weighted sum of the fair classifier $r_\lambda$ and the group classifier $g$. We run logistic regression to find $b_1, b_2 \in \mathbb{R}$ such that $\mathbb{1}(r_\lambda(x) - b_1 g(x) - b_2 > 0)$ recovers the prediction $y = \mathbb{1}(h(x) > 0)$. Coefficients $a_1$ and $a_2$, or $b_1$ and $b_2$ are found using validation data.

**Error Bars and Random Baseline** For both experiments, a substantial challenge lies in the random behavior of training a deep network. The solution found highly depends on its initial seed. To take

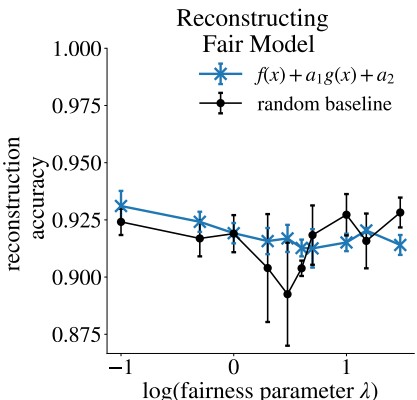 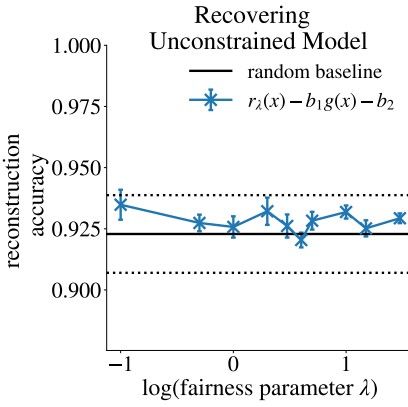

Figure 4: *Left:* **Reconstructing the fair classifier.** For a range of $\lambda$ ($x$-axis) we find $a_1$ and $a_2$ so that $f + a_1 g + a_2$ mimics the predictions of a regularized classifier $r_\lambda$. For the entire range of fairness parameters, the predictions of the regularized model are closely recovered by our approach. *Right:* **Recovering the unconstrained classifier.** We find parameters $b_1$ and $b_2$ such that $r_\lambda - b_1 g - b_2$ recovers the predictions of an unconstrained classifier $h$. From the fair model $r_\lambda$ and the group classifier $g$ we can replicate an unconstrained classifier's predictions, as accurately as another unconstrained model. Here, we predict ATTRACTIVE with protected attribute MALE. Reconstruction accuracy is directly comparable to rerunning the method with a different random seed (black lines, random baseline).

this instability into account, we provide baselines that measure how decisions vary when retraining networks. In the first experiment, we compute error bars by repeating the process for five different initial random seeds of $r_\lambda$. We compute a random baseline by comparing the five models to a sixth, differently initialized model $r_\lambda$. In the second experiment, we recover five unconstrained models trained with different initial seeds (for the error bars) and we compute a random baseline by comparing the predictions to a sixth unconstrained model.

**Results** In Figure 4 we show regularized ResNet50 models trained to predict ATTRACTIVE; the protected attribute is MALE (see Appendix D.3 for other tasks, models, and preprocessing results). The left panel evaluates how accurately our explicit approach recovers the predictions of the regularized model $r_\lambda$. We find that most of the error in recovering predictions can be attributed to classifier instability, and that retraining a classifier from scratch with a new random seed gives similar disagreement to using our reconstructed classifier. In the right panel of Figure 4, we plot the reconstruction accuracy of $r_\lambda - b_1 g - b_2$ with respect to the unconstrained classifier. By simply adding the group classifier response $g(x)$ to $r_\lambda(x)$, we obtain the predictions of the unconstrained classifier. **Compared to the baseline, we recover the unconstrained classifier responses for all $r_\lambda$ with similar fidelity to simply retraining the target classifier from scratch.**

## 5.2 Identifying Disparate Treatment in Deep Networks

We can now quantify the disparate treatment of a neural network. By exploiting our explicit estimation of the protected attribute, we can ask the counterfactual question: how would the decision have changed if the individual had belonged to the other group?

**Experimental Setup** As described in the previous subsection, we find the closest weighted sum $f + a_1 g + a_2$ of the two heads that best replicates the decisions of a given model $r_\lambda$. Then, for every individual $x$ in the test set, we replace the group classifier response $g(x)$ with the median output of the group that $x$ does *not* belong to. We evaluate how many times the prediction changes when the second head output is replaced by this counterfactual.

**Results** Figure 5 shows the proportion of individuals for whom their prediction changes. For the fairest models in the left panel, up to 30% of all individuals receive a different outcome when their second head output $g(x)$ is replaced by the median output of the other group. This is substantially more than the number of changed predictions, which we obtain when retraining with a different random seed (roughly 7% of the points). As expected, the proportion of changed predictions linearly

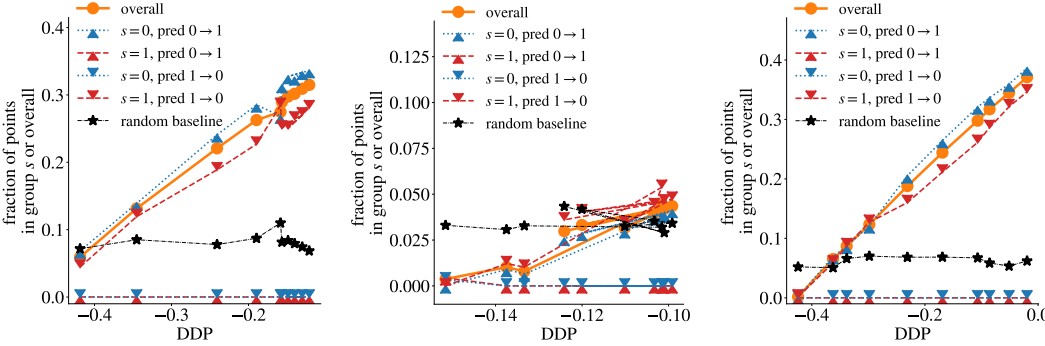

(a) ResNet50 with regularizer $\widehat{\mathcal{R}}_{\text{DP}}$ predicting ATTRACTIVE.

(b) ResNet50 with regularizer $\widehat{\mathcal{R}}_{\text{DP}}$ predicting target attribute SMILING.

(c) MobileNetV3-Small with Massaging predicting ATTRACTIVE.

Figure 5: **Uncovering disparate treatment.** How many individuals are treated differently in fair neural networks based on their protected attribute? With our approach, we find a single-headed network that approximates the decisions of the fair model (c.f. Section 5.1). Our explicit formulation allows us to replace the group classifier output $g(x)$ of an individual $x$ from group $s \in \{0, 1\}$ with the median output $\bar{g}_{1-s}$ of the other group $1 - s$. We plot the proportion of all points where the label changes (orange curves), and the proportion of points in each protected group (red and blue) for which the prediction either changed from 0 to 1 (markers pointing up) or changed from 1 to 0 (markers pointing down). *Left:* As the fairness parameter increases and fairness of the regularized model improves (zero is fair), the proportion of changed predictions increases. For the fairest model, around $30\%$ of points would obtain a different outcome if their perceived gender changed. *Center:* When we observe a small change in demographic disparity, we observe only a small proportion of points treated differently. In this plot the DDP is improved from $-0.15$ to $-0.1$. In these cases it is difficult to determine if disparate treatment is occurring. *Right:* We conduct the same experiment for *Massaging* [33]. Again, up to $35\%$ of predictions change if the predicted group membership changes.

increases with model fairness (governed by the parameter $\lambda$). Similar behavior occurs for both the regularized approach (Figure 5, left) and preprocessing (right).

While the behavior of our system is difficult to distinguish from that of a retrained fair classifier, the disagreement between retrained classifiers means that we cannot point to an individual and conclude that they received a different decision because of their protected attribute. Nonetheless, in scenarios where changing the protected attribute alters a much greater proportion of decisions than the proportion of decisions where the classifiers disagree (see Figure 5 left and right, in contrast to center) we can conclude that it is likely particular individuals suffered disparate treatment.

**Conclusion** When fair networks show the same behavior as our explicit awareness model, we can analyze the influence of group membership. Using our explicit approach, we can evaluate how fair networks systematically treat individuals differently on the basis of their protected attribute.

## 6 Legal Implications of our Analysis

We provide an extensive discussion of the legal implications of our analysis in Appendix A; here we provide an overview of our argument.

The analysis set out in the appendix is restricted to areas where the doctrine of disparate treatment is relevant. This includes areas where decisions are made concerning an individual's access to: education, employment, and housing. We start by noting that by design our approach exhibits disparate treatment. It assigns individuals into different racial or gender-based groups and uses this to alter cutoff scores in a way that explicitly violates Title VII [60] as outlined in the introduction. What is less straightforward is the relationship of the methods that we have shown to have the same systematic behavior as our new approach.

The overall legal argument can be decomposed into three parts:

1. Disparate treatment may occur even if the treatment is based on inferred attributes (such as race or sex) rather than explicitly provided attributes.

2. By construction, our explicit formulation (Section 4) exhibits disparate treatment.

3. Other fairness approaches considered exhibit the same behavior as our explicit approach, and as the relevant tests for disparate treatment are based on systematic behavior, these approaches also exhibit disparate treatment.

Finally, we explicitly identify the individuals likely disadvantaged by enforcing fairness, and consider circumstances where the use of such systems may be acceptable, and look at existing legal arguments.

## 7 Conclusion

We showed how popular existing methods for enforcing demographic parity in deep networks learn an internal representation that is more predictive of the protected attribute. The close coupling between the behavior of these approaches to our explicit model provides a tool to identify individuals who are likely to be systematically favored or disfavored by virtue of their protected attribute and to conclude that existing methods for enforcing fairness do so through disparate treatment. We have shown that the use of *some* fairness methods on *some* datasets exhibits disparate treatment. Our findings do not imply that *all* methods for enforcing demographic parity suffer from disparate treatment, merely that some can, and that caution should be used when deploying such methods.

Furthermore, our approach for enforcing fairness offers a more efficient and reliable way of enforcing a chosen degree of demographic parity. Unlike regularization-based approaches that require multiple training runs with different regularization parameters to find a desired trade-off, our approach allows for joint training of both heads once, and then a search for the desired trade-off by tuning weights using precomputed network responses on a validation set, and then finally merging the two heads. As such, it is possible to efficiently find a family of classifiers from the original architecture, of varying fairness and accuracy for little more compute than training an unfair classifier in the first place.

In hindsight, our findings are perhaps unsurprising. Regularizing methods try to balance decisions made by the classifier while minimizing the logistic loss. This requires the calibration of the classifier to be preserved as much as possible. To this end, a fairly regularized classifier should maximally distort points closest to the decision boundary, while trying to minimally alter the scores of other points. Massaging methods train one classifier, and then push points closest to the decision boundary over it, before training a new classifier to replicate these decisions. While both methods explicitly treat members of different groups differently, they do so at training time and rely on this behavior generalizing to unseen data. In contrast, our approach simply moves the decision boundary at test-time for points identified as belonging to a particular group. However, following these intuitions, it is not unreasonable that all three methods relabel the same set of points that an unconstrained classifier would be least confident about.

While our analysis presents several challenges to deploying fair ML systems in the US, it is consistent with other rulings on discrimination in US law. In general, the US requires that considerations of equity and affirmative action are satisfied by shaping an entire process to be more inclusive, and not simply by imposing race or gender-based quotas on outcomes [32, 21]. However, the opaque nature of ML makes it extremely challenging to define fair algorithms without formulating the definitions in terms of outcomes[3]. For this reason, we believe that legal reform is needed to explicitly allow the use of fair ML techniques as a tool to reduce disparate impact and increase equity.

## Acknowledgments and Disclosure of Funding

Chris Russell is also a member of the Governance of Emerging Technologies and the Trustworthy Auditing for AI project at the Oxford Internet Institute. We are grateful to other members, particularly Sandra Wachter and Johann Laux for their advice. The legal discussion strongly benefited from their thoughtful suggestions. Michael Lohaus is a member of the International Max Planck Research School for Intelligent Systems (IMPRS-IS). We thank Dominik Zietlow for his practical advice.

---

[3]See [36] for an excellent legal analysis that includes the legality of other approaches that do not explicitly rely on outcomes.

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
