# Appendix

## A Legal Implications of our Analysis

The analysis set out in this section is restricted to areas where the doctrine of disparate treatment is relevant. This includes areas where decisions are made concerning an individual's access to: education, employment, and housing. We start by noting that by design our approach exhibits disparate treatment. It assigns individuals into different racial or gender-based groups and uses this to alter cutoff scores in a way that explicitly violates Title VII [60] as outlined in the introduction. What is less straightforward is the relationship of the methods that we have shown to have the same systematic behavior as our new approach.

**Overview**

Our argument can be decomposed into three parts. We address each point in detail below:

1. Disparate treatment may occur even if the treatment is based on inferred attributes (such as race or sex) rather than explicitly provided attributes.

2. Our explicit formulation (Section 4) exhibits the behavior used to *indirectly* identify disparate treatment.

3. Other fairness approaches considered exhibit the same behavior as our explicit approach, and as the relevant tests for disparate treatment are based on systematic behavior, these approaches also exhibit disparate treatment.

Finally, we explicitly identify the individuals likely disadvantaged by enforcing fairness, and consider circumstances where the use of such systems may be acceptable, and look at existing legal arguments.

1. **Implicit Disparate Treatment** Multiple machine learning papers [76, 16, 52, 2] have asserted that machine learning systems that do not explicitly take into account knowledge of the protected attribute at prediction time cannot be performing disparate treatment. From a legal perspective, this is an oversimplification. Many of the legal tests for disparate treatment involve a demonstration of intent to treat protected groups differently [37], and it is irrelevant if knowledge of the groups is given as data from a trusted party or inferred from a photograph or other data. As an example of case law supporting this, [28] gives Hunt v. Cromartie [61] where the plaintiffs demonstrated that location was used as a proxy for race in an instance of disparate treatment. The situation considered here is even more extreme than that of Hunt v. Cromartie. As we use photographic data as input, to deny that disparate treatment can occur here is the same as denying that disparate treatment can occur on the basis of an individual's appearance.

2. **The Disparate Treatment of Our Approach** Systems which explicitly alter scoring on the basis of race or gender are widely acknowledged as being examples of disparate treatment, with [28] writing that there is such wide agreement that it is not worth discussing. While creating a system that makes explicit use of a protected attribute when making decisions demonstrates intent, it is not the only way to do so. In particular, as it is difficult to explicitly demonstrate intent when someone is either unable or unwilling to explain honestly why they made decisions, the courts recognize indirect evidence of the form: ". . . evidence, whether or not rigorously statistical, that employees similarly situated to the plaintiff other than in the characteristic (pregnancy, sex, race, or whatever) on which an employer is forbidden to base a difference in treatment received systematically better treatment" United States Court of Appeals for the Seventh Circuit [63] with similar judgments repeating these arguments occurring in ] . By design, our approach explicitly treats 'similarly situated' individuals, i.e. those who receive a similar score $f(x)$ from a classifier trained without consideration of their protected attribute, differently by changing their score and adding the term $a_1 g(x) + a_2$ which explicitly depends on their inferred protected attribute.

3. **The Disparate Treatment of Other Approaches** As the implicit argument of the previous section relies on the systematic behavior of a decision-making system, it can be directly applied to systems trained to satisfy a fairness constraint. In such cases, we only know that the system enforces the constraint, but not necessarily how. However, as we are only concerned with the system behaviour, i.e. the decisions made, if the system is closely mimicked by our approach, the same argument

applies, and we can identify those individuals with similar scores $f(x)$ who probably[4] receive different decisions by virtue of their race or gender.

**Identifying Disadvantaged Individuals**  We can therefore identify individuals that are likely to have received an unfavorable decision by virtue of their inferred protected attribute. As we can recover a close approximation of the decisions of the fair model of the form $r(x) \approx f(x) - a_1 g(x) - a_2$, individuals who initially receive a score $f(x)$ in the region $f(x) \in [a_2, a_1 + a_2)$ are likely to receive different decisions by virtue of their inferred race or gender $g(x)$.[5]

**Potential Mitigation**  One possible defense, that would only be valid in vary narrow circumstances, is to reject the relevance of the classifier $f$ trained on ground-truth data without fairness considerations, and to claim that individuals with similar $f(x)$ scores are not in fact "similarly situated" [63]. It is always possible to generate some classifier[6] $f$ from an existing classifier $r$, by subtracting any arbitrary terms of the form $a_1 g(x) + a_2$ from $r(x)$. As such, the existence of $f$ is insufficient to conclude that the existing classifier $r$ exhibits disparate treatment, and we also need to know that individuals with similar scores $f(x)$ are "similarly situated". As such, where this unaware classifier $f$ was trained on data that does not correspond to a direct measure of performance, and is known to be systematically biased [7, 67], there is little reason to believe that individuals with similar scores are also "similarly situated". This argument was proposed outside of algorithmic fairness by Selmi [57] who argued that a stronger defense could have been mounted in [62] by challenging the predictive value of the test.

**Existing Legal Arguments**

A full summary of the existing debate is out of scope for what is primarily a machine learning paper, and we only touch upon three papers to indicate the range of opinions. Kroll et al. [41] argued that protected attributes should not be used as part of the decision-making system, but that the use of ML fairness methods that use protected attributes at training time was less likely to be considered an instance of disparate treatment than an ex-post correction that explicitly alters the score of individuals with a particular race or gender. In particular, Kroll et al. [41] considered [63] and similar judgements and concluded that compared to ex-post measures[7] "incorporating nondiscrimination in the initial design of algorithms is the safest path that decision makers can take." This argument hinges upon an explicit lack of understanding of the behavior of ML systems, i.e., without knowing the details of how an opaque fair system behaves it is not possible to say that it exhibits disparate treatment. Hellman [28] argued that as disparate treatment depends upon intent, in certain circumstances, the use of protected attributes can be acceptable at decision time. Bent [9] offered two main arguments: First, that as disparate treatment hinges upon intent, the combined use of racial data (even if only at training time) with any form of fairness constraints shows an expressed intent to treat different races differently, and should also be considered disparate treatment.[8] [9]

In response to Bent [9], we note that: *(i)* bias-preserving fairness metrics [66] (i.e., the majority of existing fairness metrics) ensure that classifiers are sufficiently accurate for all groups, by matching

---

[4]As the reconstruction is not exact, we cannot be certain, however, note that the evidence provided does not even need to be *rigorously statistical* [63].

[5]This relies on $g(x)$ being close to an indicator function with $94\%$ of function responses being either 0 or 1 with a tolerance of $\pm 0.1$.

[6]Here, we consider $f$ an arbitrary classifier, and not necessarily the unconstrained classifier trained on the data.

[7]Ex-post methods adjust an already generated scoring system by choosing different thresholds for members of different groups.

[8]Bent [9] argues that this should hold for any form of fairness constraint, not just the forms of demographic parity we consider.

[9]The second argument Bent [9] considers is the difference between running an algorithm with and without race-based fairness constraints. Bent argues that any individual receiving a change in decision should be considered evidence of a difference in treatment. This argument is problematic due to the non-deterministic behavior of deep learning algorithms. As shown in Figures 4 and 5, simply rerunning the same algorithm with a different seed can result in significant changes in labels assigned to the test set, and what is needed is evidence of a systematic change in treatment over and above the expected intrinsic variability. Inherently, such evidence cannot come from considering a single individual, but must occur at the population level. See also related discussion in [36] regarding the [9] and the ill-determined nature of 'unfair' classifiers.

the distribution of errors over different groups. *(ii)* Kroll et al. [41] also argued that in [62] the court's lack of concern regarding the use of race to determine that the scoring mechanism was equally effective for all racial groups indicated that this was a legitimate use of racial data. Putting these two arguments together, it seems likely that enforcing many forms of fairness need not be a form of disparate treatment, and that it depends instead on specific facts of implementation and particularly if it is possible to identify individuals who are systematically disadvantaged by the fact of their race under a fair system.

Our position lies midway between Kroll et al. [41] and the first argument of Bent [9], and is simply that a blanket decision if algorithmic fairness violates disparate treatment is inappropriate and depends upon the facts of the particular ML system considered and the data it is deployed on. Our position is that the improved understanding from using the techniques set out in this paper is sufficient to determine that some ML fairness systems also exhibit disparate treatment (see Section 3 and Figure 1); and, more importantly, the decisions made by a fair regularized classifier are indistinguishable from those training a classifier designed to exhibit disparate treatment (see Section 4).

**Summary**

While it may not be possible to determine a priori if a particular fairness definition gives rise to disparate treatment, our decomposition of existing fair classifiers into an unconstrained classifier $f$, and a second head that rescores the response using inferred race or gender, strongly aligns with the legal definition of disparate treatment. As such, we believe that this decomposition will be of value in determining the legality of deploying fair systems in practice. From a practical perspective, this disparate treatment is most strongly observed when the unconstrained classifier $f$ has high demographic disparity, and demographic parity is strongly enforced. This makes it unlikely for any of the considered fairness methods to be appropriate tools to enforce equity without legal reform.

# B   Other Related Work

**Bias and Bias Amplification in Deep Neural Networks** Numerous papers have found deep networks discriminate based on protected groups [39, 12, 4, 22, 6] and even amplify bias present in the training data [80, 13, 70, 31, 68, 53]. Deep models have also been found to "overlearn", that is they learn representations encoding concepts that are not part of the learning objective; e.g., encoding race when trained to predict gender [59, 58]. [59] argue that overlearning is problematic from a privacy perspective as it reveals sensitive information. However, they do not consider if it allows models to disparately treat different groups. To detect unintended classifier bias, [6, 15] synthesize counterfactual images by changing latent factors of a generative model, corresponding to attributes such as race, and seeing how performance alters. This is related to our approach, however, they do not examine how fair models alter this behavior, and our decomposition of models into two heads allows us to reason counterfactually without generating images.

**Exploiting Disparate Treatment** Several papers propose to use protected information for learning group-dependent models, in order to improve performance and / or fairness, assuming that doing so is legally acceptable and the protected information is available [39, 25, 19, 64]. Also the idea of using an estimate or score of the protected attribute is not new: Menon and Williamson [49] and Oneto et al. [51] propose to infer the protected attribute from non-protected features and to use the inferred attribute to learn "group specific" models, as a way to enforce demographic parity or equalized odds. While these approaches are similar to our approach presented in Section 4, the interpretation provided by Oneto et al. is quite the opposite from ours: they consider their approach as a means of overcoming disparate treatment while we argue that such an approach should not be treated differently than an approach that explicitly uses protected information.

**Bias Mitigation Methods** In the last few years, a plethora of fairness notions, that is definitions of fairness-concerning *bias*, along with methods for mitigating such bias have been proposed, both in supervised and unsupervised learning. The methods in supervised learning are usually categorized into three groups: preprocessing methods, in-processing methods, and postprocessing methods (see [14, 48] for survey papers). In this paper we study methods from each of the three groups (cf. Section 2): the regularizer approach belongs to the group of in-processing methods (and so does our strategy proposed in Section 4), the massaging method of [33] is a preprocessing method, and the strategy of [42] is a postprocessing method. While the earlier papers on fair ML primarily considered

tabular data, more recently, bias mitigation has also been studied in the context of deep learning [17, 70, 69, 55, 71]. Fazelpour and Lipton [21] discuss a broader human-centered view going beyond altering algorithms with parity metrics.

## C Implementation Details

Here we report details omitted, for reasons of space, from the body of the paper. In Section C.1 we give details about the datasets that were used. In Section C.2, we give the formulation of a second regularizer, which is the absolute value of the relaxed DDP measure, and provide details on model training and the grid search procedure for our approach.

### C.1 Datasets

**CelebA** The CelebA dataset [43] contains $202,599$ images of celebrity faces with $40$ binary annotations, such as WEARING_GLASSES, SMILING or MALE. We use the Aligned&Cropped subset and its standard split into train, test, and validation data. We center-crop the images and resize them to $224 \times 224$. During training, we randomly crop and flip images horizontally. We use [55] as reference for choosing target and protected labels.

**FairFace** The FairFace dataset [34], published under CC BY 4.0 licence, contains $108501$ images collected from the YFCC-100M Flickr dataset and are annotated with GENDER, RACE, and AGE. We binarize the attribute RACE into WHITE and the union of all other groups. From AGE, we build several binary attributes: BELOW_20, BELOW_30, and BELOW_40. In our experiments, we use the provided validation data with 1.25 padding as our test data, and from the provided train data, we prepared our own random and balanced validation split. We center-crop the images and resize them to $224 \times 224$. During training, we crop randomly, and flip the images horizontally with probability $0.5$.

### C.2 A Second Regularizer and Experimental Details.

In Section 2 we have introduced the squared fairness regularizer $\widehat{\mathcal{R}}_{\mathrm{DP}}$ (cf. (2)), used by Wick et al. [72]. We performed our experiments also with another regularizer [47], denoted by $\widehat{\mathcal{R}}_{\mathrm{DP}}^{\mathrm{abs}}$. It is similar to $\widehat{\mathcal{R}}_{\mathrm{DP}}$, but with the squaring function replaced by the absolute value, that is

$$\widehat{\mathcal{R}}_{\mathrm{DP}}^{\mathrm{abs}}(f) := \left| \frac{1}{|\{s : s_i = 1\}|} \sum_{i \in [n]: s_i = 1} \sigma(f(x_i)) - \frac{1}{|\{s : s_i = 0\}|} \sum_{i \in [n]: s_i = 0} \sigma(f(x_i)) \right|. \quad (3)$$

**Models and Optimization** Given a fixed target attribute and protected attribute, we train all parameters of a pretrained ResNet50 [27] or MobileNetV3-Small [29] backbone provided by PyTorch with binary cross entropy loss. MobileNetV3-Small contains 2.8M parameters and is more resource friendly than the much bigger ResNet50. Hence, for some experiments we only used MobileNetV3-Small to save computation time. The dimension $m$ of the last-layer representation $z \in \mathbb{R}^m$ is $m = 2048$ for the ResNet50 and $m = 1024$ for MobileNetV3-Small.

We train all models, including our approach, with the Adam Optimizer [38] (learning rate is $10^{-4}$ on CelebA and $10^{-5}$ on FairFace, batchsize is $64$) for a total of $20$ epochs and select the model with the highest average precision achieved on the validation set. In addition, we employ a learning rate scheduler that reduces the learning rate by a factor of 10 if there is no progress on the validation loss for more than 8 epochs. To have meaningful regularizer losses for each batch, we use stratified batches, such that the prevalence of the protected attribute is roughly the same as the overall prevalence. For the classification loss, we use binary cross entropy loss with a sigmoid activation.

If we train the models with one of our two fairness regularizers, the range for the fairness parameter $\lambda$ is $\lambda \in [0, 0.1, 0.5, 1, 2, 3, 4, 5, 10, 15, 20, 30]$. For the *Massaging* preprocessing method, the range is $\lambda \in [0, 0.1, 0.2, 0.3, 0.4, 0.5, 0.6, 0.7, 0.8, 0.9, 1.0]$.

**Grid Search** The grid search procedure of our approach chooses all combinations of $a_1$ and $a_2$ from a grid of 200 equidistant points between $-15$ and $15$. Going through all combinations, we choose the most accurate model that satisfies the user-chosen fairness constraint. We continue to search in the interval of the grid points which are closest to the current solution by forming another grid of 200 equidistant points in this interval. We continue this recursion $4$ times.

# D    Extended Experimental Results

In Section D.1, we complement the results on protected attribute awareness in fair networks including results using the second regularizer (3). In Section D.2, we apply our explicit approach to MobileNetV3-Small and compare to all other fairness approaches. Finally, in Section D.3 and D.4 we extend our main result about identifying disparate treatment to different target tasks, models, and fairness approaches.

## D.1    Protected Attribute Awareness

In this section, we report further results on protected attribute awareness in fair neural networks. We plot last-layer tSNE visualizations for another CelebA task in Figure 6 and for a FairFace task in Figure 6. Similar to Figure 1 in the body of the paper, gender is separated into two clusters when we regularize the model for demographic parity.

In Figure 8, we plot the Kendall-tau correlations when using MobileNetV3-Small for both of the two presented regularizers. As with a ResNet50 model, we observe a strong association between fairness parameter and an increase in group awareness. However, for the $\widehat{\mathcal{R}}_{\mathrm{DP}}^{\mathrm{abs}}$ regularizer a positive association is less often significant than for $\widehat{\mathcal{R}}_{\mathrm{DP}}$. In Figure 9, we apply *Massaging* preprocessing with a varying fairness parameter. Results on FairFace are presented in Figures 10 and 11.

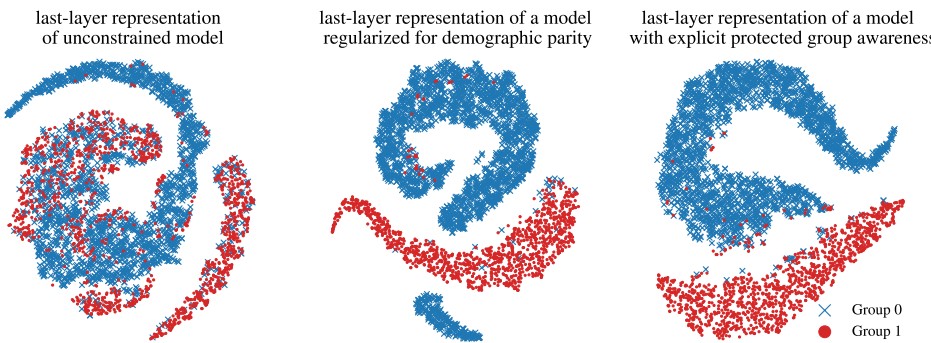

Figure 6: **tSNE [65] visualization of feature representations of unconstrained (left), fairness-regularized with $\widehat{\mathcal{R}}_{\mathrm{DP}}$ (center), and group-aware (Section 4) (right) Resnet50 models.** Each point is colored according to the protected attribute MALE, and we aim to classify the binary label ATTRACTIVE. Similar to the main paper, we observe that the fair model in the center and the group aware model on the right separate the genders.

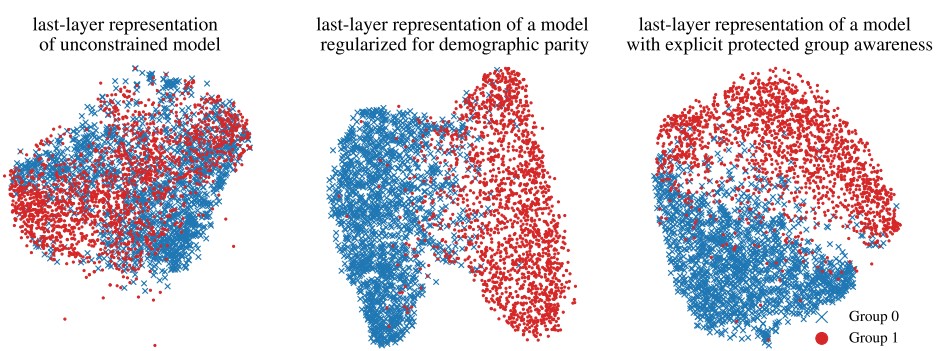

Figure 7: **tSNE visualization of feature representations of unconstrained (left), fairness-regularized (center), and group-aware (right) Resnet50 models.** In this figure, we use the Fair-Face dataset. Each point is colored according to the protected attribute GENDER, and we classify the binary label BELOW_30. Similar to CelebA, we observe that gender is separated into disjoint clusters in fair and group aware models, whereas they were mixed in the unconstrained model.

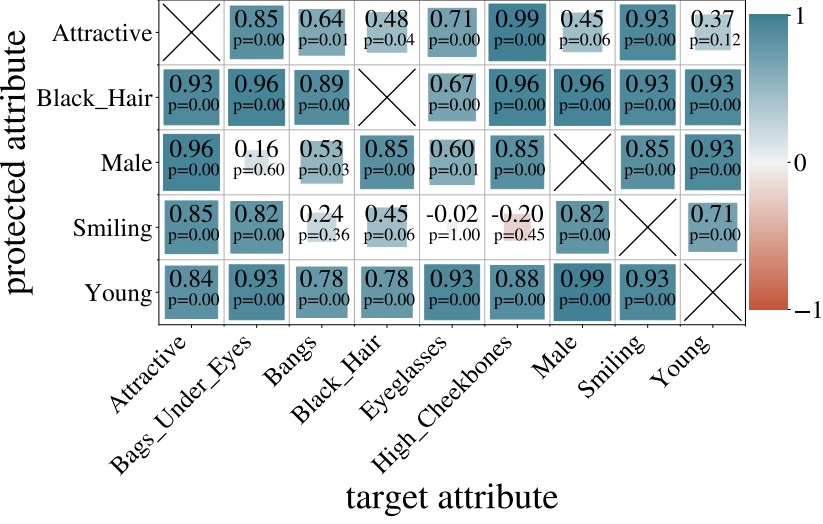

(a) MobileNetV3-Small with regularizer $\widehat{\mathcal{R}}_{\mathrm{DP}}$ on CelebA.

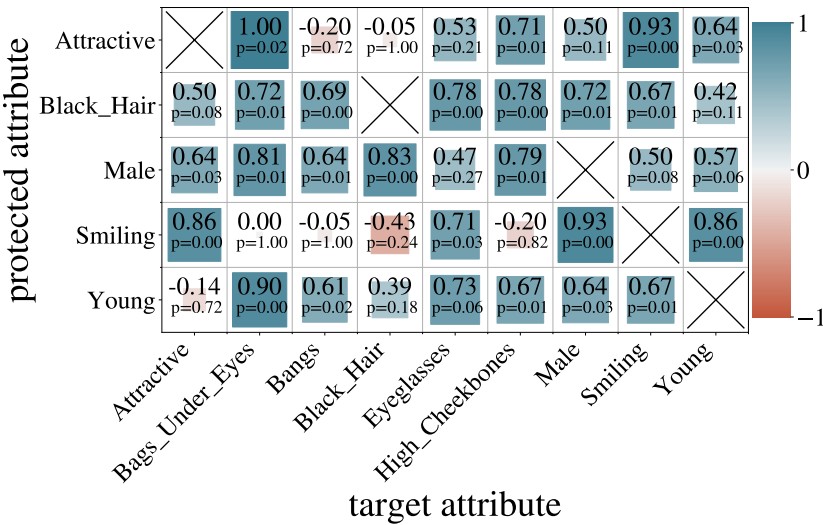

(b) MobileNetV3-Small with regularizer $\widehat{\mathcal{R}}_{\mathrm{DP}}^{\mathrm{abs}}$ on CelebA.

Figure 8: **Kendall-tau correlation between fairness parameter and protected attribute accuracy.** Similar to the results in the main paper, where ResNet50 was used, we also find for MobileNetV3-Small that group awareness is increasing as the fairness parameter is increased. In (b) we evaluate the regularizer $\widehat{\mathcal{R}}_{\mathrm{DP}}^{\mathrm{abs}}$ and, although on fewer tasks, observe a similar behavior.

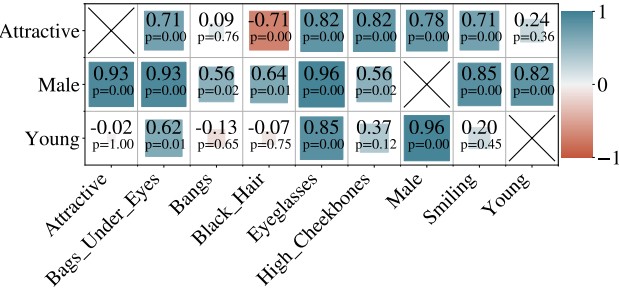

(a) MobileNetV3-Small with *Massaging* preprocessing on CelebA.

Figure 9: **Kendall-tau correlation between fairness parameter $\lambda$ and protected attribute accuracy.** Similar to the regularized approaches, we find an increased group awareness for the *Massaging* preprocessing method, especially when the protected attribute is MALE.

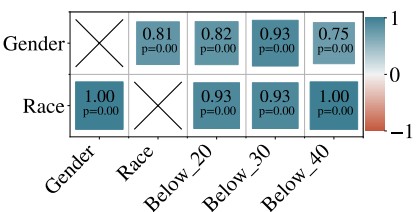

(a) MobileNetV3-Small with regularizer $\widehat{\mathcal{R}}_{\text{DP}}$ on FairFace.

(b) MobileNetV3-Small with regularizer $\widehat{\mathcal{R}}_{\text{DP}}^{\text{abs}}$ on FairFace.

Figure 10: **Kendall-tau correlation between fairness parameter $\lambda$ and protected attribute accuracy.** Similar to the findings on the CelebA dataset, we also find an increased group awareness on FairFace for the protected attributes RACE and GENDER.

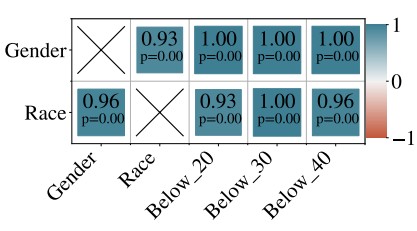

(a) ResNet50 with regularizer $\widehat{\mathcal{R}}_{\text{DP}}$ on FairFace.

(b) ResNet50 with regularizer $\widehat{\mathcal{R}}_{\text{DP}}$ on FairFace.

Figure 11: (Left) **Kendall-tau correlation between fairness parameter $\lambda$ and protected attribute accuracy.** (Right) **Increase of protected attribute accuracy** of the group classifier learned on the last layer of ResNet50.

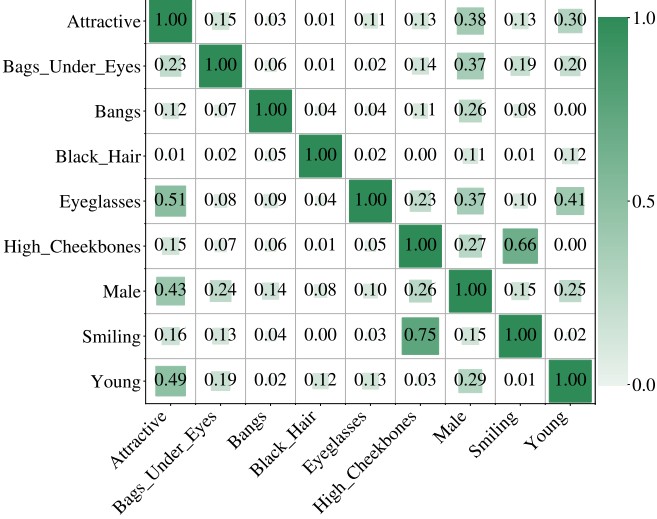

(a) Demographic parity violation (DDP) of unconstrained ResNet50 on CelebA.

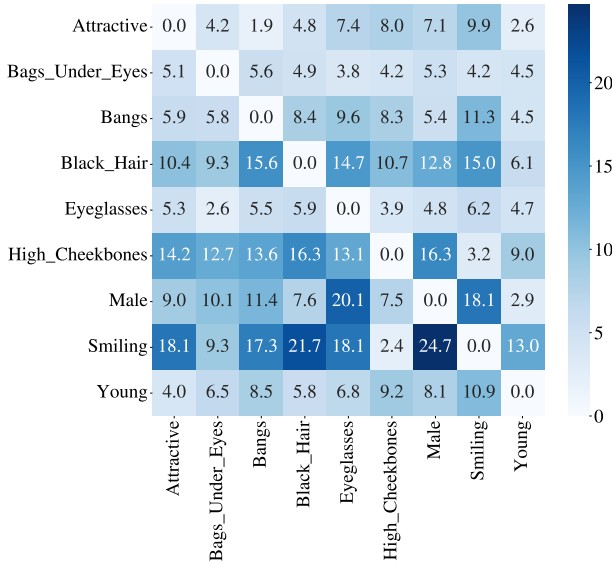

(b) Maximum increase of protected attribute accuracy when training ResNet50 with regularizer $\widehat{\mathcal{R}}_{\mathrm{DP}}$ on CelebA.

Figure 12: (Top) **Demographic parity violation (DDP) of the unconstrained classifier.** The increase in group awareness is more moderate for those tasks where the unconstrained classifier is very unfair, for example for the task (column) MALE. However, this is not always the case as for protected attribute SMILING and target BANGS for example. (Bottom) **Maximum increase of protected attribute accuracy.** Compared to the unconstrained model, we show the highest difference to the second head accuracy of fair models. Even though the unconstrained model is fair, for example for a few target tasks when protected attribute is SMILING, the increase in second head accuracy can still be large.

### D.2 Our Explicit Approach.

Table 2: **Accuracy under strict fairness constraints.** The first block requires a reduction of the absolute value of the DDP of at least 50%, the second at least 80%. The protected attribute is MALE from CelebA dataset and we use the MobileNetV3-Small architecture. Crosses indicate that the method did not achieve the required fairness. Similar to the main paper, the regularizer $\widehat{\mathcal{R}}_{\mathrm{DP}}$ often fails to find sufficiently fair solution. The regularizer $\widehat{\mathcal{R}}_{\mathrm{DP}}^{\mathrm{abs}}$ always finds fair solutions, however, at high costs in accuracy, often resulting in trivial solutions. **Our explicit two-headed approach can always find a fair solution** and is comparable to Lipton, which, contrary to us, requires the true protected attribute.

| | Attractive | Bags_Under_Eyes | Bangs | Black_Hair | Eyeglasses | High_Cheekbones | Smiling | Young |
|---|---|---|---|---|---|---|---|---|
| | | | 50% disparity reduction | | | | | |
| Lipton | 0.8034 | 0.8441 | 0.9473 | 0.9001 | 0.9658 | 0.8609 | 0.9200 | 0.8773 |
| Our Approach | 0.8023 | 0.8439 | 0.9445 | 0.8930 | 0.9773 | 0.8613 | 0.9212 | 0.8762 |
| Massaging | 0.7986 | 0.8424 | 0.9396 | 0.9012 | 0.9661 | 0.8572 | 0.9135 | 0.8520 |
| Regularizer $\widehat{\mathcal{R}}_{\mathrm{DP}}$ | 0.7959 | 0.8446 | ✗ | 0.8993 | ✗ | 0.8603 | ✗ | 0.8710 |
| Regularizer $\widehat{\mathcal{R}}_{\mathrm{DP}}^{\mathrm{abs}}$ | 0.7935 | 0.8311 | 0.8443 | 0.8962 | 0.9545 | 0.8609 | 0.4997 | 0.8728 |
| | | | 80% disparity reduction | | | | | |
| Lipton | 0.7775 | 0.8352 | 0.9331 | 0.9001 | 0.9618 | 0.8403 | 0.9106 | 0.8606 |
| Our Approach | 0.7741 | 0.8347 | 0.9310 | 0.8921 | 0.9652 | 0.8443 | 0.9105 | 0.8580 |
| Massaging | 0.7612 | 0.8204 | ✗ | 0.8947 | ✗ | ✗ | ✗ | 0.8468 |
| Regularizer $\widehat{\mathcal{R}}_{\mathrm{DP}}$ | 0.7740 | 0.8294 | ✗ | 0.8989 | ✗ | ✗ | ✗ | 0.8562 |
| Regularizer $\widehat{\mathcal{R}}_{\mathrm{DP}}^{\mathrm{abs}}$ | 0.7693 | 0.8311 | 0.8443 | 0.8962 | 0.9407 | 0.5182 | 0.4997 | 0.8307 |

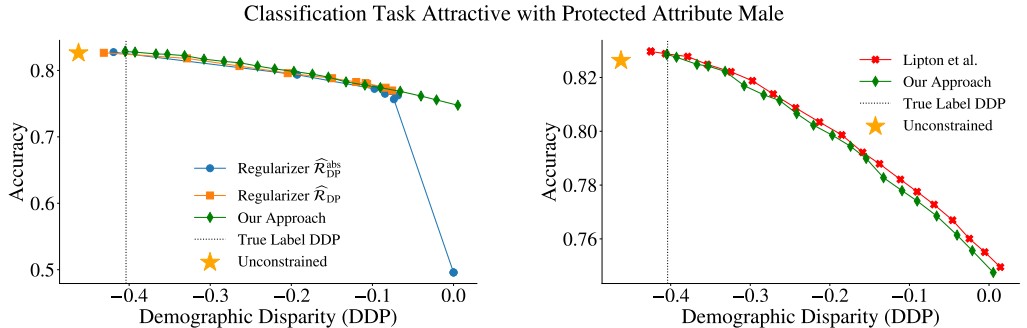

Figure 13: **Comparison of different fairness approaches using the MobileNetV3-Small architecture.** We compare our group aware model to fairness-regularized models (*left plot*) and the approach of Lipton et al. [42] (*right plot*) on when predicting the target ATTRACTIVE with respect to the protected attribute MALE. For all methods, we observe the typical trade-off: as the model becomes fairer (DDP is closer to 0), the target accuracy for ATTRACTIVE decreases. All methods obtain similar accuracy for a particular DDP value. However, the regularizer $\widehat{\mathcal{R}}_{\mathrm{DP}}$ is unable to achieve near perfect fairness and saturates around a DDP value of $-0.1$. The regularizer $\widehat{\mathcal{R}}_{\mathrm{DP}}^{\mathrm{abs}}$ collapses to a trivial fair solution. Note that Lipton et al. [42] requires the protected attribute at test time, while we infer the protected attribute.

## D.3 Fair Networks Behave like our Explicit Approach.

In this section, we conduct the experiments from Section 5.1 on other tasks and computer vision models. Predicting SMILING we use our approach to reconstruct fair models and recover the unconstrained model using a ResNet50 with $\widehat{\mathcal{R}}_{\mathrm{DP}}$ regularizer (Figure 14), using a MobileNetV3-Small with $\widehat{\mathcal{R}}_{\mathrm{DP}}$ regularizer (Figure 15), and using a MobileNetV3-Small with *Massaging* preprocessing (Figure 16). In Figure 17 and 18, we recover the unconstrained model from fair ResNet50 and MobileNetV3-Small models predicting either ATTRACTIVE or YOUNG. Overall, we are able to replicate the behavior of fair models using both heads of our explicit approach and to recover the unconstrained model from a fair model with the group classifier head. Sometimes, as observed in Figure 17 the unconstrained classifier cannot be recovered from the fairest models within the performance of the random baseline.

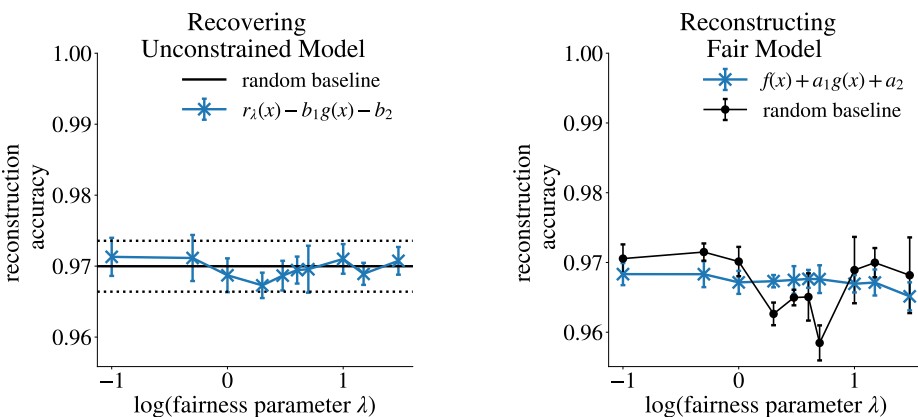

Figure 14: **Recovering the unconstrained classifier and reconstructing fair classifiers.** We train ResNet50 models with the $\widehat{\mathcal{R}}_{\mathrm{DP}}$ regularizer for the target SMILING and protected attribute MALE. Again, we can reconstruct fair models with our approach and we can recover the unconstrained model by adding the second head to the fair model.

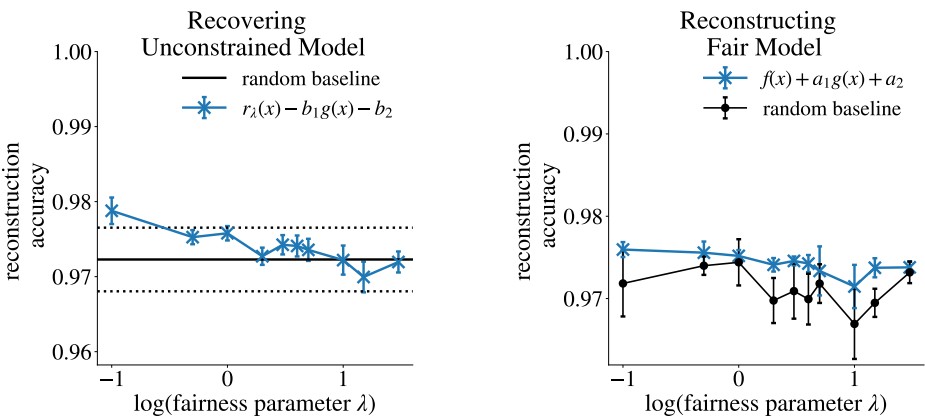

Figure 15: **Recovering the unconstrained classifier and reconstructing fair classifiers.** We train MobileNetV3-Small models with the $\widehat{\mathcal{R}}_{\mathrm{DP}}$ regularizer for the target SMILING and protected attribute MALE. Similarly to the analysis above with a ResNet50, our observations hold for MobileNetV3-Small models as well.

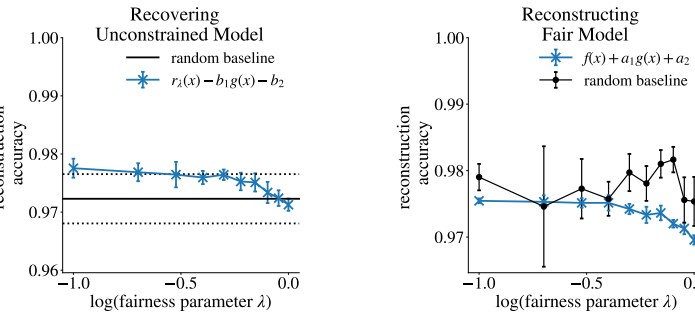

Figure 16: **Recovering the unconstrained classifier and reconstructing fair classifiers.** We train MobileNetV3-Small models with the *Massaging* preprocessing for the target SMILING and protected attribute MALE. When using *Massaging*, we can reconstruct the resulting fair models with our approach. However, it reconstructs the most fair models slightly less accurately than a retrained fair model.

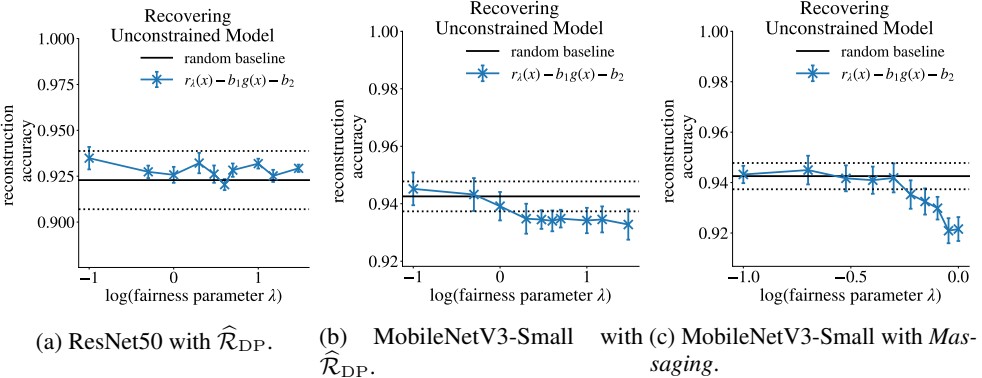

(a) ResNet50 with $\widehat{\mathcal{R}}_{\mathrm{DP}}$.  (b) MobileNetV3-Small with $\widehat{\mathcal{R}}_{\mathrm{DP}}$.  (c) MobileNetV3-Small with *Massaging*.

Figure 17: **Recovering the unconstrained classifier.** For different models and fairness approaches for the target ATTRACTIVE and protected attribute MALE, we evaluate how our approach can reproduce the behavior of the fair model. From regularized ResNet50 models we can recover the unconstrained model well. From fair regularized or massaged MobileNetV3-Small models we recover the unconstrained model slightly worse than a retrained unconstrained model.

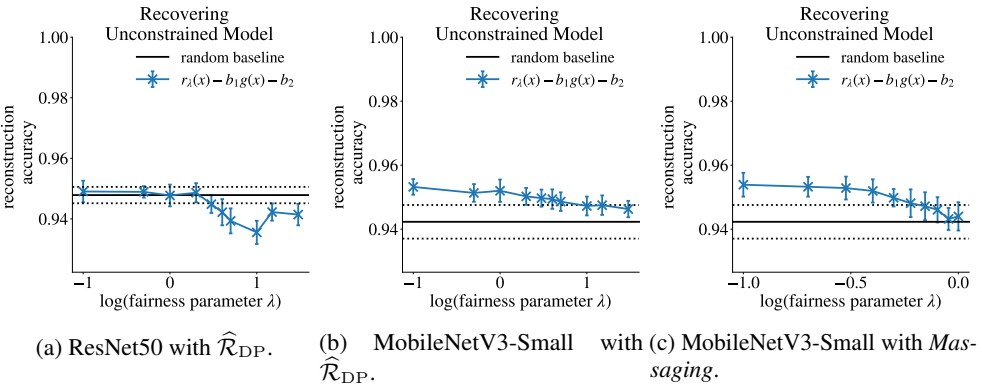

(a) ResNet50 with $\widehat{\mathcal{R}}_{\mathrm{DP}}$.  (b) MobileNetV3-Small with $\widehat{\mathcal{R}}_{\mathrm{DP}}$.  (c) MobileNetV3-Small with *Massaging*.

Figure 18: **Evaluating reconstruction accuracy of our approach.** For different models and fairness approaches for the target YOUNG and protected attribute MALE, we evaluate how close our approach is. In this example, we can recover the unconstrained model from fair models using the second head well. The accuracy for fair ResNet50 models is below the random baseline.

## D.4 Disparate Treatment in Fair Networks.

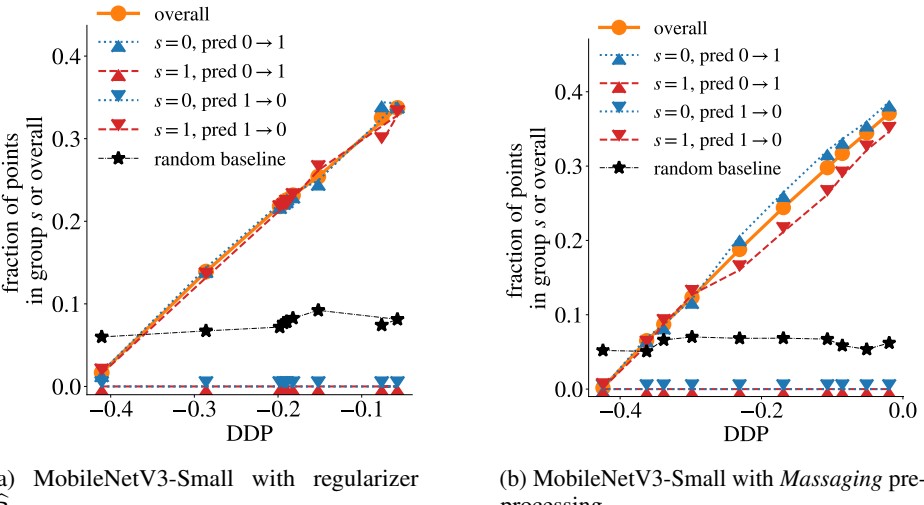

(a) MobileNetV3-Small with regularizer $\widehat{\mathcal{R}}_{\mathrm{DP}}$.

(b) MobileNetV3-Small with *Massaging* pre-processing.

Figure 19: We perform our analysis described in Section 5.2 on CelebA with target attribute AT-TRACTIVE and protected attribute MALE. For both the regularizer and the preprocessing, up to $35\%$ of all points would receive a different outcome if their inferred attribute changed. As expected, only in one group negative predictions change into positive predictions; at the same time only for the other group positive prediction change to negative predictions.

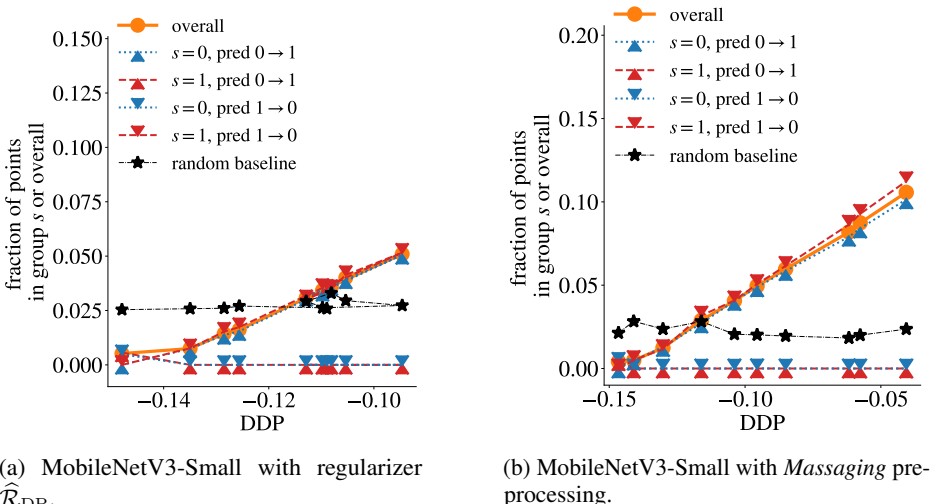

(a) MobileNetV3-Small with regularizer $\widehat{\mathcal{R}}_{\mathrm{DP}}$.

(b) MobileNetV3-Small with *Massaging* pre-processing.

Figure 20: We perform our analysis described in Section 5.2 on CelebA with target attribute SMIL-ING and protected attribute MALE. We trained MobileNetV3-Small models with (Left) the $\widehat{\mathcal{R}}_{\mathrm{DP}}$ regularizer, or (Right) the *Massaging* preprocessing method. We observe a similar behavior as above. Since the regularized approach is not improving fairness further than $\mathrm{DDP} = -0.1$, the fraction of changed predictions is only slightly higher than the baseline.