# OpenReview forum: "Are Two Heads the Same as One? Identifying Disparate Treatment in Fair Neural Networks"
_NeurIPS.cc/2022/Conference — NeurIPS 2022 Accept_

### Official Review · Reviewer_yjGc · 2022-07-10

**Rating:** 6
**Confidence:** 4
**Soundness:** 2 fair
**Presentation:** 3 good
**Contribution:** 3 good

**Summary:**

The paper observes that the fair training model may use sensitive group information in the internal state of the network. Such a phenomenon may be undesirable, as fair training fundamentally aims to not use group information. To support this main claim, the paper gives several interesting observations; for example, the fairer model can predict the groups more accurately. Also, the paper proposes a two-headed model training strategy that can demonstrate the disparate treatment of fair training. The main experimental results are based on fairness regularizer and massaging techniques, which represent fairness in-processing and pre-processing, respectively. The paper also connects this observation with a meaning of fairness from the legality point of view.

**Questions:**

Q1: As briefly mentioned in the previous section, the in-depth discussion on the observations seems not enough.

- First, it would be very helpful if the paper gives any intuition on why some fair training models use group information to achieve fairness.
- Also, regularizer and massaging are quite different techniques, but there is no explanation on 1) how similar phenomena are observed in these different techniques or 2) if there is any different trend between the two methods. Since regularizer is an in-processing method, it will actively guide the model to consider fairness during the training. Thus, it is a bit imaginable why the regularized method may result in undesirable information usage. However, massaging is a pre-processing method, so it is not clear why the fixed training data make a similar effect on the model. It would be great if the paper clearly explain such details.

Q2: It seems various details on the two-headed model are missing.

- It would be helpful if the paper gives a rationale for the loss function design. For example, the loss function in Section 4 contains two separate terms (i.e., L_bce and g loss), but there is no tuning knob to adjust the impact between the two terms. Since the roles of the two terms are completely different, I was wondering whether we do not need any tuning knob in that loss.

- Also, why the two-headed model is a post-processing method? I understand that the method can adjust the a_1 and a_2 values of the trained model to achieve the target (fairness) performances. However, it seems this method initially needs model training by using the loss in Section 4. Thus, I believe that the paper needs to clarify the category of their methods. I am not saying that the paper should change the category, but a more careful representation is needed.


**Limitations:**

The paper discusses the limitations and potential negative societal impacts.

**Strengths And Weaknesses:**

<Strengths>

- The main observation that the fair training model may rather use sensitive group information is very interesting. Figures 1 and 2 show these results well. As many fair algorithms have been proposed, I believe such an underlying analysis is now important to further extend model fairness literature.
- The idea of the two-headed model is simple and intuitive, and it is reasonable to illustrate the paper’s claims. Using the unfair model to explain the undesirable property of the fair algorithms gives new viewpoints.

<Weaknesses>

Although I appreciate their observations and enjoyed reading this paper, there are also several weak aspects in the paper.

- The paper contains various phenomena, but the in-depth discussions to give intuitions on the phenomena are not enough. Detailed questions are in Q1 in the next section.
- Several details in the two-headed model are not clearly explained. Detailed questions are in Q2 in the next section.

---

> ### Author Response · Authors · 2022-08-01
> **Answers to Questions 1 and 2**
>
> We thank the reviewer for the constructive feedback and the positive comments. We are glad that you enjoyed reading our paper. In the following, we clarify remaining questions raised.
>
> **Q1**
> Unfortunately, text discussing this was cut due to space constraints. We will add it to the Appendix. For your convenience we reproduce it below:
>
> "In hindsight, our findings are perhaps unsurprising. Regularizing and Massaging can be seen as altering  the labels assigned to individuals in the training set on the basis of their protected attribute. Regularized approaches do so by enforcing a soft quota, while Massaging explicitly relabels individuals. As the trained network generalizes from training to test data it brings this behavior with it. Still, it requires the comparison to our  approach to confirm these intuitions and to demonstrate that disparate treatment of this kind not only happens on training data, but also on test data."
>
> To expand further, we can informally consider the behavior of all three methods on the training set:
> 1) Regularized approaches try to balance the decisions made by the classifier while minimizing the logistic loss. This requires them to preserve as much as possible the calibration of the classifier, and they achieve this by distorting the behavior on points closest to the decision boundary, while trying to keep other points intact.
> 2) Preprocessing trains one classifier, and then takes points closest to the decision boundary, and pushes them over the boundary, before training a new classifier to replicate these decisions.
> 3) Our approach simply moves the decision boundary for points identified as belonging to a particular protected group.
> As such, it is not unreasonable that all three methods relabel the same set of points that they are least confident about.
>
> **Q2**
> Extra tuning parameters. Typically for high-capacity models, the training error goes to zero for both terms, and only limited improvement can come from reweighting tasks. While it is possible that we could have obtained an epsilon improvement in performance by further tuning, we did not attempt this.
>
> Post-processing. Yes, our method jointly trains a second head, and as such, does require alteration of the training procedure. We consider it a post-processing method as it doesn't induce a fair classifier until we merge the two heads in a single post processing step. We clarified line 163. Note that similar ambiguity occurs elsewhere in the literature, for example, the preprocessing method of [1] alters the data after first training an initial classifier.
>
> [1] Faisal Kamiran and Toon Calders. Data preprocessing techniques for classification without discrimination. Knowledge and Information Systems (KAIS), 2012.

---

> > ### Comment · Reviewer_yjGc · 2022-08-08
> > **After reading the authors' responses**
> >
> > I appreciate the authors’ response. Most of the responses are reasonable to me. Thus, I would increase my score to 6 (weak accept).
> >
> > In addition to the authors' response, I hope that I can see careful explanations in their revision.
> > 1) When the authors explain that regularizing, massaging, and their algorithm have something in common, I would recommend that the paper also clarify again that such a common aspect may not apply to other fairness algorithms.
> > 2) If the paper only assumes the high-capacity models, this should also be clearly stated, since fairness literature is studied in various datasets and model types.
> >
> > The reason that I give this recommendation is that the paper's observations may lead to hasty generalizations. I know that the authors explained that their observations are not applied to all fair algorithms in the conclusion section. However, I think such a clarification would be better to appear in earlier sections as well (e.g., Introduction).

---

> > > ### Author Response · Authors · 2022-08-08
> > > **Thanks**
> > >
> > > Yes, we fully agree.
> > > As we were explaining this, I had the strong feeling that the intuition behind these methods was important and shouldn't be left out of the main body of the paper.
> > >
> > > If accepted we will use the extra space in the camera ready to explain the properties of these methods more clearly.

---

### Official Review · Reviewer_qBCV · 2022-07-11

**Rating:** 3
**Confidence:** 4
**Soundness:** 2 fair
**Presentation:** 3 good
**Contribution:** 2 fair

**Summary:**

This paper is largely an observational paper, with a particular implementation of a two-headed network, which conducts experiments related to the fairness of classifiers. Specifically, the endeavor to conceptualize whether “fair” classification systems may violate US equal protection laws, like protection against disparate treatment or impact. These authors look at a network trained with a subgroup performance difference regularizer and observe that the more a fairness metric is enforced, the easier it is for the protected attribute of an intermediate representation of a data point to be predicted. They propose a two-headed classification task to create classifiers which adhere to some demographic parity constraints. Consequently, they believe that a fair model can be approximated by their approach which leads them to conclude that fair models exhibit disparate treatment and thus may violate US equal protection laws.


**Questions:**

1. I see the authors include as a reference this paper (https://arxiv.org/pdf/1707.00075.pdf, http://proceedings.mlr.press/v80/madras18a/madras18a.pdf) but do not discuss it in their main text.
2. Why do the authors omit any codebase or output data in their submission?
3. Can the authors describe how \lambda is used? It is merely described as a fairness parameter in the paper and the appendix does not provide any more clarity. Is this like the Lagrangian so minimize the loss + \lambda \hat{R}_{DP}(f)?
4. In Section 3, why do you care about the rank of the \lambdas and the overall accuracy? i.e., why perform a rank sum test (ordinality) and not just a Spearman’s rank test?
5. I’m further not convinced by Section 3 given the very large class imbalances that generally exist in these protected attribute categories. Can you speak to how these class imbalances impact your results? Two other cofounders are (1) the overall performance of the ResNet model, and (2) the co-occurance of the protected and the predicted variables. Can you update your statistical analysis to account for these through regression or anova?
6. I’d appreciate more clarity in the main body of the paper on the definition of g in Lines 150-151. I feel this is an important enough topic the reader shouldn’t have to search in the Appendix for this.
7. Can you speak about the sensitivity of the a_1, a_2, b_1, and b_2 values that are chosen by the logistic regression in Section 5.1? I’m concerned about how these values would fare under new datapoint.

Typo: penultimate line of Figure 4’s caption: “iMale”. There may be others.

**Limitations:**

The authors have adequately addressed the limitations and potential negative societal impact of their work.


**Strengths And Weaknesses:**

Strengths
1. The topics very important the the authors take an interesting and thought-provoking position on the legality of these approaches.
2. Section 4 proposes and demonstrates an approach with compelling results.
3. The paper is written clearly and the authors take a methodical approach to arguing their conclusions.

Weaknesses
1. The literature review in this paper should be improved. For example, since the authors are only discussing only some of the algorithmic approaches to address mistreatment in classification, the authors should broader place the regularization and preprocessing approaches within the larger algorithmic fairness work. Additionally, the authors do not describe other two-headed approaches to fair classification like (https://arxiv.org/pdf/1707.00075.pdf, http://proceedings.mlr.press/v80/madras18a/madras18a.pdf)
2. I have some methodological concerns which I have questions for in the next section. Once I have clarity on these, I’ll be better able to evaluate the conclusions of your paper.
3. It appears that this approach is only looking at binary protected attributes and would be hard to extend to multi-category labels.
4. The claims made in the beginning of the paper about “extensive results” could be reasonably argued for Section 3, but become increasingly less convincing in Sections 4 and 5. For instance, the conclusion drawn in Section 5.1 is for just one protected attribute and predicted variable pair.

---

> ### Author Response · Authors · 2022-08-01
> **Addressing questions and weaknesses**
>
> We thank the reviewer for the effort put into this detailed review and we hope to address the issues raised.
>
> Since in the reviewer’s opinion we take 'an interesting and thought-provoking position on the legality of these approaches', we would first like to emphasize that, to the best of our knowledge, our paper is the first to provide rigorous evidence of disparate treatment in ML systems that do not require the protected attribute at test time. Others have argued that using the protected attribute only during training but not during testing does not constitute disparate treatment. We have shown that a fair neural network trained this way behaves like a model that uses the inferred protected attribute to construct group-dependent decision thresholds, a clear case of disparate treatment.
> We agree with the reviewer that the topic is very important and we ask to reconsider the reject rating, given that they say the paper is both sound and well-presented. Particularly, if this score is based on issues such as missing code and references, which are easily resolved and we are happy to fix for the final version.
>
> We first address what the reviewer mentioned as weaknesses and then respond to the reviewer‘s questions:
>
> Weaknesses:
> 1. Lit review. Yes, space is strongly constrained in the paper and there are many related works we could discuss. A more extended literature review is presented in Appendix B that discusses some of these works. The adversarial approaches are very different, and train an adversarial second head in a minmax formulation to remove information about the protected group, while the objectives of our two heads are jointly minimized in a multitask setting. However, we will discuss the adversarial second head of representation learning approaches more clearly in Appendix B.
>
> 2. Addressed in detail later.
>
> 3. Extending this to a multigroup setting is straightforward, and we make use of this in currently unpublished work. Instead of predicting [0,1] in a second head, we can learn a multiclass head to predict a one-hot encoding of the groups, still minimizing the least squares loss. Then we tune a vector of offsets o such that $f(x) + o\cdot g(x)) > 0$ optimizes for demographic parity.
>
> 4. We were unable to find this exact quote, if the reviewer can give us a line number, we can clarify the text. Note that we are presenting a method for testing if behavior is consistent with disparate treatment, and demonstrating that disparate treatment can occur in practice. We do not claim to provide enough evidence to show that it always occurs --  after all it will not occur in every case (for example, if the existing classifier is already fair) but it is something that can and should be checked using the analysis we developed in this paper.
> We also extend the results of Section 5.1 in Appendix D.3 by predicting Young and Smiling. While we do not consider as many target attribute-protected attribute pairs as in Section 3, these experiments require the training of hundreds of models. To save resources, we focused on three attributes, but extended our results by choosing two different neural networks (ResNet and MobileNet) and two different fairness methods (regularizer and Massaging).  Our observations hold across models, methods, and target attributes.
>
> Questions:
> 1. See weakness 1.
>
> 2. The principal contribution of our work is not the codebase, but an analysis of disparate treatment in ML systems and the legal implications. However, we are happy to release code of our two-headed approach with the final paper.
>
> 3. Yes, lambda is a trade-off parameter. Higher values of lambda encourage an increase in fairness at the expense of accuracy. We clarified line 83.
>
> 4. We do not expect group awareness to have a linear relationship with lambda, only a monotonic one (when we increase lambda by a fixed amount, group accuracy need not increase by the same amount, but it does go up). As such a ranking correlation is appropriate. The two choices are Kendall's or Spearman's. However, Kendall's rank correlation is known to be more robust and better suited for smaller sample sizes [1].
>
> 5. It's possible you have misread our analysis. We fix the dataset and backbone, and then explicitly set lambda to a range of values. For each choice of dataset and backbone we report the ranked correlation.  Because we are setting lambda to the same range of values each time, no confounding is possible. We are happy to respond to messages about this and clarify further.
>
> 6. Thanks, we clarified line 152/153.
>
> 7. Stability on unseen data. In our experiments, weights are selected on validation data and we compare the predictions of the fair network to predictions of our weighted sum on **unseen test data**.
> Typo: Fixed. Thanks.
>
> [1] Croux, C. and Dehon, C. (2010). Influence functions of the Spearman and Kendall correlation measures. Statistical Methods and Applications, 19, 497-515.

---

> > ### Author Response · Authors · 2022-08-02
> > **Ad Question 4 - Spearman's correlation**
> >
> > Regarding Question 4, we have recreated Figure 2 using the Spearman's correlation suggested by the reviewer. The results can be found under https://imgur.com/a/zRKXTXn
> > As expected the results are close to identical to Kendall's and we find for almost all tasks that the accuracy of predicting the protected attribute increases as we increase the fairness parameter lambda.

---

> ### Author Response · Authors · 2022-08-05
> **Any follow-up questions?**
>
> I hope that we have addressed all the issues raised to your satisfaction.
>
> As you said you would reevaluate on the basis of our answers, can you please ask any follow-up questions now, while we are still able to answer them.
>
> Thank you for your time.

---

### Official Review · Reviewer_Qj8s · 2022-07-12

**Rating:** 6
**Confidence:** 4
**Ethics Flag:** Yes
**Soundness:** 2 fair
**Presentation:** 2 fair
**Contribution:** 3 good

**Summary:**

The paper shows how research on methods to enforce demographic parity can be more predictive of protected attributes. Considering the legal implications of disparate treatment, the work merges a second classifier that explicitly predicts the protected attribute. The addition of this second classifier helps further identify individuals systematically disadvantaged.

**Questions:**

The legal discussion seems almost disjointed from the other work. It seems like the motivation for the work stems from the legal discussion, and while the legal implications nicely complement the results, how does the approach presented contribute to the presented legal analysis?

**Ethics Review Area:**

["Discrimination / Bias / Fairness Concerns"]

**Limitations:**

The authors discuss limitations of scope and only having tested on a few datasets.

**Strengths And Weaknesses:**

The authors provide a simple yet novel addition to the fairness literature. And the work is properly positioned in the literature.

---

> ### Author Response · Authors · 2022-08-01
> **Our approach is key to our legal analysis**
>
> Thank you for your comments and the positive feedback on our paper.
>
> The legal discussion explicitly builds off the work in the remainder of the paper. Key to the argument is the fact that the legal identification of disparate treatment can be behavioral. Our network exhibits disparate treatment by design, and this allows us to argue that where other methods for fairness exhibit the same behavior (i.e. the difference between our method and others is similar to retraining with a new random seed), this can be said to be disparate treatment.
>
> For fairness methods that do not exhibit the same behavior, we cannot identify disparate treatment using our argument. In particular, our analysis does not directly apply to other methods with different accuracy-fairness trade-offs to the methods we consider. They do not share the same behavior, and cannot be said to violate our test for disparate treatment.

---

### Review · Ethics_Reviewer_8x3X · 2022-07-29

**Recommendation:** No changes are needed.

**Ethics Review:**

A primary objective of this paper is to discuss the legal implications of algorithmic fairness methodology in the context of disparate treatment. While this paper discusses ethical issues, it does not raise its own ethical issues of concern.

---

### Meta-Review · Area_Chair_hrzB · 2022-08-30

**Recommendation:** Accept
**Confidence:** Less certain

**Metareview:**

This paper identifies and addresses an interesting phenomenon in fair ML, where "fairer" networks are in turn more discriminatory in a different sense.  The paper is an insightful read and, while I don't agree that prior literature has bee done due diligence (please see reviews/discussion), I do believe the work stands alone as a solid contribution.  I would ask the authors to continue to pay attention to Reviewer qBCV, who unfortunately did not participate in the discussion session, but with whom I do agree; many of their points are valid and should be addressed either in the camera ready for this conference or in a future version of the work.

**Award:**

No

---

### Decision · Program_Chairs · 2022-09-14

Accept